# TAN WITHOUT A BURN: SCALING LAWS OF DP-SGD

## ABSTRACT

Differentially Private methods for training Deep Neural Networks (DNNs) have progressed recently, in particular with the use of massive batches and aggregated data augmentations for a large number of steps. These techniques require much more compute than their non-private counterparts, shifting the traditional privacy-accuracy trade-off to a privacy-accuracy-compute trade-off and making hyper-parameter search virtually impossible for realistic scenarios. In this work, we decouple privacy analysis and experimental behavior of noisy training to explore the trade-off with minimal computational requirements. We first use the tools of Rényi Differential Privacy (RDP) to show that the privacy budget, when not overcharged, only depends on the total amount of noise (TAN) injected throughout training. We then derive scaling laws for training models with DP-SGD to optimize hyper-parameters with more than a $100\times$ reduction in computational budget. We apply the proposed method on CIFAR-10 and ImageNet and, in particular, strongly improve the state-of-the-art on ImageNet with a $+9$ points gain in accuracy for a privacy budget $\varepsilon = 8$.

## 1 INTRODUCTION

Deep neural networks (DNNs) have become a fundamental tool of modern artificial intelligence, producing cutting-edge performance in many domains such as computer vision (He et al., 2016), natural language processing (Devlin et al., 2018) or speech recognition (Amodei et al., 2016). The performance of these models generally increases with their training data size (Brown et al., 2020; Rae et al., 2021; Ramesh et al., 2022), which encourages the inclusion of more data in the model's training set. This phenomenon also introduces a potential privacy risk for data that gets incorporated. Indeed, AI models not only learn about general statistics or trends of their training data distribution (such as grammar for language models), but also remember verbatim information about individual points (e.g., credit card numbers), which compromises their privacy (Carlini et al., 2019; 2021). Access to a trained model thus potentially leaks information about its training data.

The gold standard of disclosure control for individual information is Differential Privacy (DP) (Dwork et al., 2006). Informally, DP ensures that the training algorithm does not produce very different models if a sample is added or removed from the dataset. Motivated by applications in deep learning, DP-SGD (Abadi et al., 2016) is an adaptation of Stochastic Gradient Descent (SGD) that clips individual gradients and adds Gaussian noise to their sum. Its DP guarantees depend on the privacy parameters: the sampling rate $q = B/N$ (where $B$ is the batch size and $N$ is the number of training examples), the number of gradient steps $S$, and the noise variance $\sigma^2$.

Training neural networks with DP-SGD has seen progress recently, due to several factors. The first is the use of pre-trained models, with DP finetuning on downstream tasks (Li et al., 2021; De et al., 2022). This circumvents the traditional problems of DP, because the model learns meaningful features from public data and can adapt to downstream data with minimal information. In the remainder of this paper, we only consider models trained *from scratch*, as we focus on obtaining information through the DP channel. Another emerging trend among DP practitioners is to use massive batch sizes at a large number of steps to achieve a better tradeoff between privacy and utility: Anil et al. (2021) have successfully pre-trained BERT with DP-SGD using batch sizes of 2 million. This paradigm makes training models computationally intensive and hyper-parameter (HP) search effectively impractical for realistic datasets and architectures.

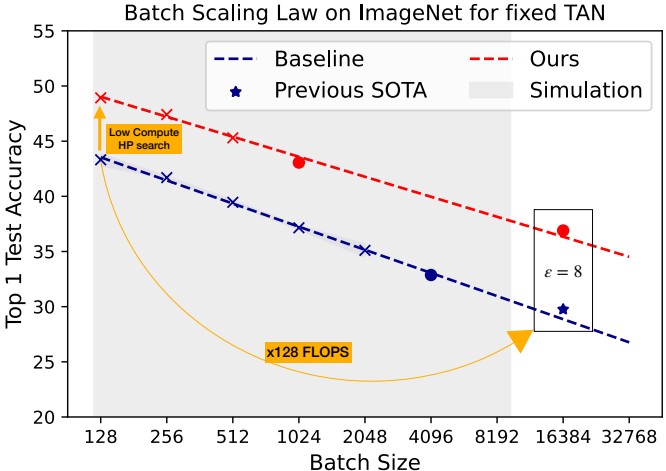

Figure 1: Training with DP-SGD on ImageNet for a constant number of steps $S = 72k$. All points are obtained at constant $\sigma/B$, with $\sigma_{\text{ref}} = 2.5$ and $B_{\text{ref}} = 16384$. The dashed lines are computed using a linear regression on the crosses, and the dots and stars illustrate the predictive power of TAN. We perform low compute hyper-parameter (HP) search at batch size $128$ and extrapolate our best setup for a single run at large batch size: stars show our reproduction of the previous SOTA from De et al. (2022) and improved performance obtained under the privacy budget $\varepsilon = 8$ with a $+6$ points gain in top-1 accuracy. The shaded blue areas denote 2 standard deviations over three runs.

In this context, we look at DP-SGD through the lens of the Total Amount of Noise (TAN) injected during training, and use it to decouple two aspects: privacy accounting and influence of noisy updates on the training dynamics. We first show that within a wide range of the privacy parameters, the privacy budget $\varepsilon$ is a function only of the total amount of noise. Using the tools of RDP accounting, we approximate $\varepsilon$ by a closed-form expression. We then analyze the scaling laws of DNNs at constant TAN and show that performance at very large batch sizes (computationally intensive) is (linearly) predictable from performance at small batch sizes as illustrated in Figure 1.

In summary, our contributions are as follows:

- We define the notion of Total Amount of Noise (TAN) and show that when the budget $\varepsilon$ is not overcharged, it only depends on TAN;

- We derive scaling laws and showcase the predictive power of TAN to reduce the computational cost of hyper-parameter tuning with DP-SGD, saving a factor of $128$ in compute on ImageNet experiments (Figure 1). We then use TAN to find optimal privacy parameters, leading to a gain of $+9$ points under $\varepsilon = 8$ compared to the previous SOTA;

- We leverage TAN to quantify the impact of the dataset size on the privacy/utility trade-off and demonstrate that doubling dataset size halves $\varepsilon$ while providing better performance.

## 2 BACKGROUND AND RELATED WORK

In this section, we review traditional definitions of DP and RDP. We consider a randomized mechanism $\mathcal{M}$ that takes as input a dataset $D$ and outputs a machine learning model $\theta \sim \mathcal{M}(D)$.

**Definition 1** (Differential Privacy). *A randomized mechanism $\mathcal{M}$ satisfies $(\varepsilon, \delta)$-DP (Dwork et al., 2006) if, for any pair of datasets $D$ and $D'$ that differ by one sample and for all subset $R \subset \mathbf{Im}(\mathcal{M})$,*

$$\mathbb{P}(\mathcal{M}(D) \in R) \leq \mathbb{P}(\mathcal{M}(D') \in R) \exp(\varepsilon) + \delta. \tag{1}$$

DP-SGD (Abadi et al., 2016) is the most popular DP algorithm to train DNNs. It selects samples uniformly at random with probability $q = B/N$ (Poisson sampling), clips per-sample gradients to a norm $C$ ($\text{clip}_C$), aggregates them and adds (Gaussian) noise. With $\theta$ the parameters of the DNN:

$$g_{\text{noisy}} = \frac{1}{B} \sum_{i \in B} \text{clip}_C \left( \nabla_\theta \ell(\theta, (x_i, y_i)) \right) + \mathcal{N} \left( 0, \frac{C^2 \sigma^2}{B^2} \right). \tag{2}$$

The traditional privacy analysis of DP-SGD is obtained through Rényi DP.

**Definition 2** (Rényi Divergence). *For two probability distributions $P$ and $Q$ defined over $\mathcal{R}$, the Rényi divergence of order $\alpha > 1$ of $P$ given $Q$ is:*

$$D_\alpha(P \parallel Q) := \frac{1}{\alpha - 1} \log \mathbb{E}_{x \sim Q} \left( \frac{P(x)}{Q(x)} \right)^\alpha.$$

**Definition 3** (Rényi DP). *A randomized mechanism $\mathcal{M} \colon \mathcal{D} \to \mathcal{R}$ satisfies $(\alpha, d_\alpha)$-Rényi differential privacy (RDP) if, for any $D, D \in \mathcal{D}'$ that differ by one sample, we have*

$$D_\alpha(\mathcal{M}(D) \parallel \mathcal{M}(D')) \leq d_\alpha.$$

Rényi DP is a convenient notion to track privacy because composition is additive: a sequence of two algorithms satisfying $(\alpha, d_\alpha)$ and $(\alpha, d'_\alpha)$ RDP satisfies $(\alpha, d_\alpha + d'_\alpha)$ RDP. In particular, the succession of $S$ steps of a $(\alpha, d_\alpha)$ RDP mechanism satisfies $(\alpha, Sd_\alpha)$ RDP. Mironov et al. (2019) show that each step of DP-SGD satisfies $(\alpha, g_\alpha(\sigma, q))$-RDP with

$$g_\alpha(\sigma, q) := D_\alpha((1 - q)\mathcal{N}(0, \sigma^2) + q\mathcal{N}(1, \sigma^2) \parallel \mathcal{N}(0, \sigma^2)).$$

Finally, a mechanism satisfying $(\alpha, d_\alpha)$-Rényi-DP also satisfies $(\tilde{\varepsilon}, \delta)$-DP (Mironov, 2017) for $\tilde{\varepsilon} = d_\alpha + \frac{\log(1/\delta)}{\alpha - 1}$. Training for $S$ steps with DP-SGD thus satisfies $(\varepsilon_{RDP}, \delta)$-DP with

$$\varepsilon_{\mathrm{RDP}} := \min_\alpha Sg_\alpha(\sigma, q) + \frac{\log(1/\delta)}{\alpha - 1}. \tag{3}$$

RDP is the traditional tool used to analyse DP-SGD, but other accounting tools have been proposed to obtain tighter bounds (Gopi et al., 2021). In this work, we use the accountant due to Balle et al. (2020), whose output is referred to as $\varepsilon$, which is slightly smaller than $\varepsilon_{\mathrm{RDP}}$.

**Training from Scratch with DP-SGD.** Training ML models with DP-SGD typically incurs a loss of model utility, and recent work have shown that using very large batch sizes improves the privacy/utility trade-off (Anil et al., 2021; Li et al., 2021). De et al. (2022) recently introduced Augmentation Multiplicity (AugMult), which averages the gradients from different augmented versions of every sample before clipping and leads to improved performance on CIFAR-10. Computing persample gradients with mega batch sizes for a large number of steps and AugMult makes DP-SGD much more computationally intensive than non-private training, typically dozens of times. For instance, reproducing the previous SOTA on ImageNet of De et al. (2022) under $\varepsilon = 8$ necessitates a 4-day run using 32 A100 GPUs, while the non-private SOTA can be reproduced in a few hours with the same hardware (Goyal et al., 2017). Yu et al. (2021b) propose to use low-rank reparametrization of the weight matrices and diminish the computational cost of accessing per-sample gradients.

**Finetuning with DP-SGD.** Tramer & Boneh (2020) shows that handcrafted features are very competitive when training from scratch, but fine-tuning deep models outperforms them. Li et al. (2021); Yu et al. (2021a) fine-tune language models to competitive accuracy on several NLP tasks. De et al. (2022) consider models pre-trained on JFT-300M and transferred to downstream tasks.

## 3 THE TAN APPROACH

We introduce the notion of Total Amount of Noise (TAN) and discuss its connections to RDP accounting. We then demonstrate how training with reference privacy parameters $(q_{\mathrm{ref}}, \sigma_{\mathrm{ref}}, S)$ can be simulated with much lower computational resources using the same TAN with a smaller batch size.

**Definition 4.** *Let the individual signal-to-noise ratio $\eta$ (and its inverse $\Sigma$, the Total Amount of Noise or TAN) be as follows:*

$$\eta^2 = \frac{1}{\Sigma^2} := \frac{q^2 S}{2\sigma^2}.$$

### 3.1 MOTIVATION

We begin with a simple case to motivate our definition of TAN. We assume a one-dimensional model, where the gradients of all points are clipped to $C$. Looking at Equation 2, in one batch, the

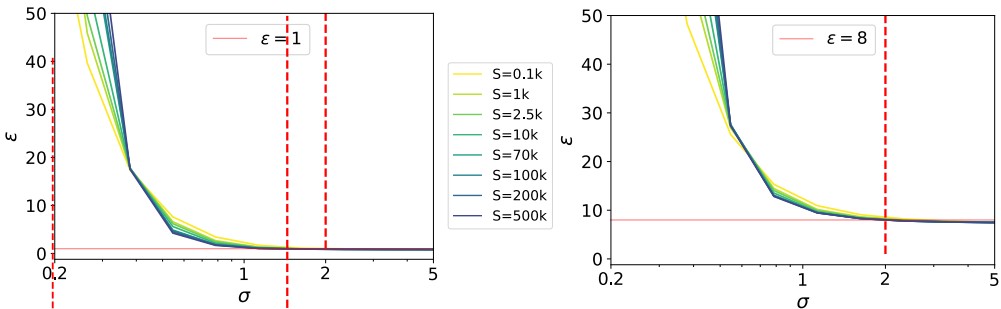

Figure 2: Privacy budget $\varepsilon$ as a function of the noise level $\sigma$ with $\eta$ constant. On both figures, each curve corresponds to a different number of steps $S$, and each point on the curve is computed at a sampling rate $q$ such that $\eta$ is constant. On the left, we use $\eta = 0.13$ (resulting in $\varepsilon_{\text{TAN}} = 1$ in Equation 4). On the right, we use $\eta = 0.95$ ($\varepsilon_{\text{TAN}} = 8$). We observe a "privacy wall" imposing $\sigma \geq 0.5$ for meaningful level of privacy budget $\varepsilon$, and $\sigma \geq 2$ for constant $\varepsilon \approx \varepsilon_{\text{TAN}}$.

expected signal from each sample is $C/B$ with probability $q = B/N$ and $0$ otherwise. Therefore, the expected individual signal of each sample after $S$ steps is $SC/N$, and its squared norm is $S^2C^2/N^2$. The noise at each step being drawn independently, the variance across $S$ steps adds to $SC^2\sigma^2/B^2$. The ratio between the signal and noise is thus equal to (up to a factor $1/2$)

$$\frac{\frac{S^2C^2}{N^2}}{\frac{2SC^2\sigma^2}{B^2}} = \frac{q^2S}{2\sigma^2} = \eta^2.$$

Denoting $\eta_{\text{step}} := q/\sqrt{2}\sigma$, we have $\eta^2 = S\eta_{\text{step}}^2$. The ratio $\sigma/q$ is noted by Li et al. (2021) as the effective noise. The authors found that for a fixed budget $\varepsilon$ and fixed $S$, the effective noise decreases with $B$. Our analysis goes further by analyzing how RDP accounting explains this dependency.

## 3.2 CONNECTION WITH PRIVACY ACCOUNTING

Intuitively, we expect that the privacy budget $\varepsilon$ only depends on the signal-to-noise ratio $\eta$. In Figure 2, we plot $\varepsilon$ as a function of $\sigma$ and $S$, at a *fixed* $\eta$, and observe that $\varepsilon$ is indeed constant, but only when $\sigma > 2$. On the contrary, when $\sigma$ gets smaller, $\varepsilon$ increases exponentially, creating a "Privacy Wall". We can shed light on this phenomenon by looking at the underlying RDP values. We observe (see Figure 3) that when $\sigma > 2$, $g_\alpha(\sigma, q)$ is close to $\alpha q^2/(2\sigma^2) = \alpha\eta_{\text{step}}^2$. Replacing that in the definition of $\varepsilon_{\text{RDP}}$ (Equation 3), we get

$$\varepsilon_{\text{RDP}} \approx \eta^2 + \min_\alpha \left( (\alpha - 1)\eta^2 + \frac{\log(1/\delta)}{\alpha - 1} \right) = \eta^2 + 2\eta\sqrt{\log(1/\delta)} =: \varepsilon_{\text{TAN}}(\eta). \qquad (4)$$

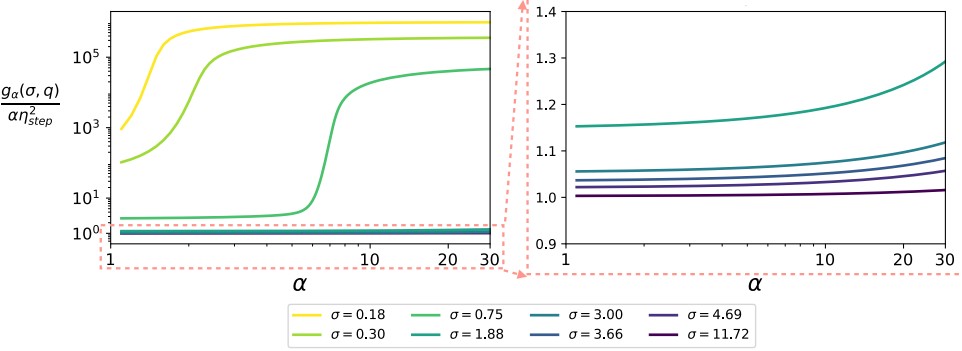

Figure 3: Approximation of $g_\alpha(\sigma, q)$. All curves correspond to distinct couples $(q, \sigma)$ such that $\eta_{\text{step}} = 3.9 \times 10^{-3}$ (used for ImageNet). The right plot corresponds to an enlargement of the left plot: the ratio is very close to $1$ for $\sigma \geq 2$. The phase transitions on the left plot were also observed by Wang et al. (2018) and Abadi et al. (2016).

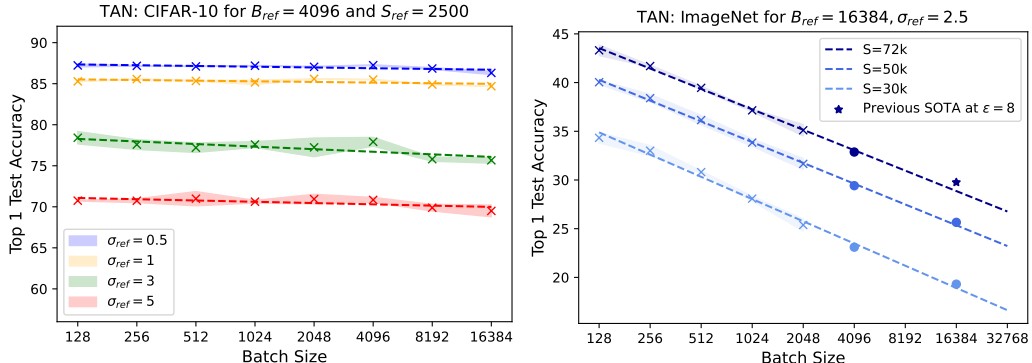

Figure 4: Test accuracies at constant $\eta_{\text{step}} = B_{ref}/(\sqrt{2}N\sigma_{ref})$ and $S$ are (log) linearly decreasing with $B$. Dashed lines are computed using a linear regression on the crosses. Shaded areas correspond to 3 std confidence intervals. (Left) CIFAR-10 with 16-4-WideResNet for $S_{ref}$ steps. Each curve corresponds to a different value of $\eta_{\text{step}}$. (Right) ImageNet with NF-ResNet-50 with various numbers of steps. The scaling law holds for various training configurations.

We verify this relationship empirically, and in particular choose $\eta$ to get a desired $\varepsilon_{\text{TAN}}$ in Figure 2. Conversely, we observe that when $\sigma < 2$, $g_\alpha$ becomes much larger than $\alpha\eta_{\text{step}}^2$, which explains the blow-up in $\varepsilon$ from Figure 2. Having this simple approximation for $\varepsilon$ is useful in practice because it allows for simple mental gymnastics: for instance, doubling the sampling rate $q$ while dividing the number of steps $S$ by 4 should leave the privacy budget constant, which we observe empirically.

## 3.3 SCALING AT CONSTANT TAN

Our RDP analysis in Section 3.2 suggests a simple scaling strategy. Starting from $(q, \sigma, S)$, while $\sigma < 2$, we can double both $q$ and $\sigma$. This drastically improves privacy accounting (Figure 2), and the per step signal-to-noise ratio $\eta_{\text{step}}$ remains constant. However, given that the number of steps $S$ is fixed, each doubling of $q$ doubles the computational cost.

**Batch Scaling Laws.** We now analyse how this strategy affects the performance of the network. In Figure 4, we perform this analysis on CIFAR-10 and ImageNet. We find that for triplets $(q, \sigma, S)$ for which $q/\sigma = q_{\text{ref}}/\sigma_{\text{ref}}$ (keeping $\eta_{\text{step}}$ constant), the performance of the network (log) linearly decreases with the batch size. This is consistent with the (non-private) work of Smith et al. (2020), which shows that for a fixed number of steps, small batch sizes perform better in generalisation (i.e., test accuracy) than large batch sizes. The difference with our work is that we consider noisy updates and observe a (log) linear relationship between batch size and performance.

**Choice of $\sigma$.** If $\sigma < 2$, simultaneously doubling $q$ and $\sigma$ has a small or negligible negative impact on accuracy (Figure 4) but it can greatly reduce the privacy budget (Figure 2). Reciprocally, halving $\sigma$ and $q$ is slightly beneficial or neutral to the performance (Figure 4), and if $\sigma > 4$, it keeps the privacy guarantees *almost* unchanged (Figure 2). It also divides the computational cost by 2. This explains why state-of-the-art approaches heuristically find that mega-batches work well: a blind grid search on the batch size and the noise level at constant privacy budget is likely to discover batches large enough to have $\sigma > 2$. Our analysis gives a principled explanation for the sweet spot of $\sigma \in [2, 4]$ used by most state-of-the-art approaches (De et al., 2022; Li et al., 2021).

**Efficient TAN training.** We go one step further and study training in the small batch size setting. We choose the optimal hyper-parameters (including architecture, optimizer, type of data augmentation) in this simulated setting, and finally launch one single run at the reference (large) batch size, with desired privacy guarantees. On ImageNet, we target $B_{\text{ref}} = 16{,}384$ with $B = 128$. We thus reduce the computational requirements by a factor of $128\times$. Each hyper-parameter search in the ImageNet setup takes 4 days using 32 A-100 GPUs; we reduce it to less than a day on a single GPU.

Table 1: Top-1 test accuracy when training on ImageNet from scratch using DP-SGD. We train a NF-ResNet-50 with $\sigma = 2.5$, hyper-parameters of Table 2 and $(B, S) = (32768, 18k)$ (Table 4).

| Method | $(\varepsilon, \delta)$ | Accuracy |
|---|---|---|
| Kurakin et al. (2022) | $(13.2, 10^{-6})$ | 6.2% |
| De et al. (2022) (original paper) | $(8, 8.10^{-7})$ | 32.4% |
| De et al. (2022) (our reproduction) | $(8, 8.10^{-7})$ | 30.2% |
| Ours | $(8, 8.10^{-7})$ | **39.2%** |

## 4 EXPERIMENTS

We leverage our efficient TAN training strategy and obtain new state-of-the-art results on ImageNet for $\varepsilon = 8$ (Table 1). We then study the impact of the dataset size on the pricacy/utility trade-off. We also demonstrate how our low compute simulation framework can be used to detect performance bottlenecks when training with noisy updates: in our case, the importance of the order between activation and normalization in a WideResNet on CIFAR-10.

### 4.1 EXPERIMENTAL SETUP

We use the CIFAR-10 dataset (Krizhevsky et al., 2009) which contains 50K $32 \times 32$ images grouped in 10 classes. The ImageNet dataset (Deng et al., 2009; Russakovsky et al., 2014) contains 1.2 million images partitioned into 1000 categories. When using data augmentation, we always use Augmentation Multiplicity as detailed in Appendix C. For both datasets, we train models from random initialization. On CIFAR-10, we train 16-4-WideResNets (Zagoruyko & Komodakis, 2016). On Imagenet, we compare Vision Transformers (ViTs) (Dosovitskiy et al., 2020), Residual Neural Networks (ResNets) (He et al., 2016) and Normalizer-Free ResNets (NF-ResNets) (Brock et al., 2021b). We fix $\delta = 1/N$ where $N$ is the number of samples for each experiment and report the corresponding value of $\varepsilon$. We use $C = 1$ for the clipping factor in Equation 2 as we did not see any improvement using other values. We use the Opacus (Yousefpour et al., 2021) library in Pytorch (Paszke et al., 2019). We open-source the training code.

### 4.2 IMAGENET

In Section 4.2.1, we use our simulated training with constant TAN to find optimal hyperparameters at low compute and improve performance for reference privacy parameters $(B_{\text{ref}}, \sigma_{\text{ref}}, S) = (16384, 2.5, 72k)$. In Section 4.2.2, we find optimal privacy parameters at constant TAN by changing the number of steps $S$, leading to a new state of the art on ImageNet for $\varepsilon = 8$ (Table 1).

#### 4.2.1 HYPER-PARAMETER TUNING AT CONSTANT TAN

**Hyper-parameter search.** We run a large hyper-parameter search and report the best hyper-parameters in Table 2 as well as the corresponding improvement for various batch sizes (at constant

Table 2: Comparing optimal hyper-parameters. Keeping $\eta_{step}$ and $S$ constant, we compare various changes in the training pipeline. We compare with the baseline of De et al. (2022) (blue line in Figure 1: NFResNet-50, learning rate at 4, EMA decay at 0.99999, 4 random augmentations averaged over 3 runs). Each gain is compared to the previous column.

| | Imagenet: $\sigma_{\text{ref}} = 2.5$, $B_{\text{ref}} = 16{,}384$, $S = 72K$ | | | | | | |
|---|---|---|---|---|---|---|---|
| $B$ | $(lr, \mu, d)$ | | decay | | AugMult | | AugTest | Total |
| 128 | $(8, 0, 0)$ | +1.0 | 0.999 | +1.2 | (Ours, 8) | +3.0 | +0.4 | +5.6% |
| 256 | $(8, 0, 0)$ | +0.8 | 0.999 | +1.2 | (Ours, 8) | +3.0 | +0.7 | +5.7% |
| 512 | $(8, 0, 0)$ | +1.2 | 0.999 | +1.1 | (Ours, 8) | +2.8 | +1.1 | +6.2% |
| 1024 | $(8, 0, 0)$ | +1.6 | 0.999 | +1.2 | (Ours, 8) | +2.3 | +0.8 | +5.9 % |
| 16384 | - | - | - | - | - | - | +0.8 | **+6.7%** |

Table 3: Low compute simulation of privacy parameter search. We start from $B = 256 = 16384/64$ and $S = 72$K. We use $\sigma = 2.5/64$ for all runs and no data augmentation.

| $B_{\text{ref}} = 256, S_{\text{ref}} = 72$K | | | |
|---|---|---|---|
| $S$ | $B$ | $lr$ | Gain |
| 9K | 756 | 64 | -6.22% |
| 18K | 512 | 32 | **+1.32%** |
| 72K | 256 | 8 | / |
| 288K | 128 | 2 | -1.88% |

$\eta_{\text{step}}$ and $S$). Each gain is compared to the optimal hyper-parameters find at the previous column. We search over learning rates $lr \in [1, 2, 4, 8, 12, 16]$, momentum parameters $\mu \in [0, 0.1, 0.5, 0.9, 1]$ and dampening factors $d \in [0, 0.1, 0.5, 0.9, 1]$. We use exponential moving average (EMA) on the weights (Tan & Le, 2019) with a decay parameter in $[0.9, 0.99, 0.999, 0.9999, 0.99999]$.

We try different types of data augmentation, that we referred to as "RRC", "Ours" and "SimCLR", and try for each various multiplicity of augmentations $(1, 2, 4, 8, 16)$ (see Appendix C for details).

- RRC: a standard random resized crop (crop chosen at random with an area between $8\%$ and $100\%$ of the original image and random aspect ratio in $[3/4, 4/3]$),
- Ours: random crop around the center with 20 pixels padding with reflect, random horizontal flip and color jitter;
- SimCLR: the augmentation from Chen et al. (2020), including color jitter, grayscale, gaussian blur and random resized crop, horizontal flip.

We find (Table 2) that optimal parameters are the same in each scenario of simulation, as predicted in Section 3.3. We perform one run with these optimal parameters at $B = 16384$ which satisfies a privacy budget of $\varepsilon = 8$. Note that we use multiple batch sizes only to support our hypothesis and batch scaling law, but it is sufficient to simulate only at $B = 128$. Our experiments indicate that AugMult is the most beneficial when the corresponding image augmentations are rather mild.

**Testing with augmentations.** We also test the model using a majority vote on the augmentations of each test image (AugTest column in Table 2). We use the same type and number of augmentations as in training. It improves the final top-1 test accuracy. This is in line with a recent line of work aiming at reconciling train and test modalities (Touvron et al., 2019). To provide a fair comparison with the state of the art, we decide **not to** include this gain in the final report in Table 1 and Table 4.

**Choice of architecture and optimizer.** We have experimented with different architectures (ViTs, ResNets, NFResnets) and optimizers (DP-Adam, DP-AdamW, DP-SGD) (see Appendix B for details). Our best results are obtained using a NFResnet-50 and DP-SGD with constant learning rate, which differs from standard practice in non-private training.

### 4.2.2 Privacy parameter search at constant TAN

While we kept $S$ constant in previous experiments, we now explore constant TAN triplets $(q, \sigma, S)$ by varying $S$. We keep $\sigma$ fixed to 2.5 and vary $(B, S)$ starting from the reference $(16384, 72$K$)$ at

Table 4: Privacy parameter search. We use the optimal parameters described in Section 4.2.1 with $\sigma = 2.5$ for one expensive run and compare it with our optimal result

| $B_{\text{ref}} = 16384, S_{\text{ref}} = 72$K | | | | |
|---|---|---|---|---|
| $S$ | $B$ | $\varepsilon$ | $lr$ | Test acc |
| 18K | 32,768 | 8.00 | 32 | **39.2%** |
| 72K | 16,384 | 7.97 | 8 | 36.1% |

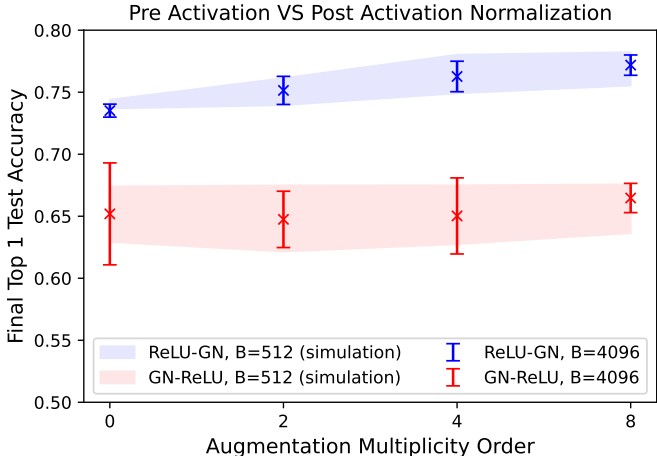

Figure 5: Impact of data augmentation on the test accuracy using pre-activation normalization vs post-activation in a 16-4-WideResnet on CIFAR-10. We compare simulation at $(B, \sigma) = (512, \frac{3}{8})$ and reference $(B_{\text{ref}}, \sigma_{\text{ref}}) = (4096, 3)$, both trained for $S = 2{,}500$ steps. Confidence intervals are plotted with two standard deviations over 5 runs. Augmentation Multiplicity Order corresponds to the number of augmentations per image, or $K$ in Appendix C.

constant $\eta = q^2 S/(2\sigma^2)$. Given that $\sigma > 2$, we stay in the *almost* constant privacy regime (see Figure 2): we indeed observe $\varepsilon \approx \varepsilon_{\text{TAN}}$ in Table 4. We scale the learning rate inversely to $S$ to compensate for the decrease of the noisy updates' magnitude (Equation 2). Since performing this privacy parameter search is computationally intensive, we first simulate training using our scaling law at $B = 256$ (with the same $\eta_{\text{step}}$) and display our results in Table 3. Our best results are obtained for 18k steps. Finally, we perform one computationally expensive run at $S = 18$k and $B = 32768$, with other hyper-parameters from Section 4.2.1, and show the results in Table 4.

We note an improvement over our previous best performance at $(B, \sigma, S) = (16384, 2.5, 72\text{K})$ referred in Table 2. Overall, we improved performance by $9\%$ when training from scratch on ImageNet with DP-SGD under $\varepsilon = 8$. We compare to our reproduction of the previous SOTA of De et al. (2022) at $30.2\%$ (compared to the results reported in the original paper ($32.4\%$), we still gain $7\%$ of accuracy). Thus, we have shown how we can use TAN to perform optimal privacy parameter search while simulating each choice of optimal parameters at a much smaller cost.

### 4.3 ABLATION

We now illustrate the benefit of TAN for ablation analysis. We study the importance of the order between activation and the normalization layers when training with DP-SGD. We also discuss how gathering more training data helps improve performance while decreasing the privacy budget. On both experiments, we train a 16-4-WideResnet on CIFAR-10, constant learning rate at $4$, and we are studying $(B_{\text{ref}}, \sigma_{\text{ref}}, S) = (4096, 3, 2.5\text{k})$ (reference private training).

**Pre-activation vs Post-activation Normalization**  Normalization techniques such as BatchNorm (Ioffe & Szegedy, 2015), GroupNorm (GN) (Wu & He, 2018) or LayerNorm (Ba et al., 2016) help training DNNs. Note that BatchNorm is not compatible with DP-SGD because it is not amenable to per-sample gradient computations, we thus resort to GroupNorm. These normalization layers are usually placed between convolutional layers and activations (e.g., CONV-GN-ReLU). Brock et al. (2021a) suggest that signal propagation improves when the order is reversed (to CONV-ReLU-GN).

We experiment with DP-SGD training using both orders of layers, and display our results in Figure 5. We make two observations. First, the reverse order leads to significantly greater performance, and is more robust. Second, the standard order does not benefit from data augmentation. We observe that the two simulated experiments with $B = 512$ represented by lighter colors in Figure 5 (2 standard deviations around the means) have the same properties. However, each simulation is $8$ times less

Table 5: Impact of the training set size $N$ on the privacy/utility trade-off. We start training on 10% of the data ($N_0 = 5K$). We use $B = 4,096$, $\sigma = 3$ and $S = 2,500$, with post-activation normalization, and no augmentation. Standard deviations are computed over 3 runs.

| CIFAR-10: $\sigma = 3$, $B = 4,096$, $S = 2,500$ | | |
|---|---|---|
| $N$ | $\varepsilon$ | Test acc (%) |
| 5K | 150.3 | 59.9 ($\pm$1) |
| 25K | 13.7 | 71.1 ($\pm$0.4) |
| 40K | 7.3 | 72.9($\pm$0.1) |
| 50K | 7.1 | 74.0 ($\pm$0.5) |

computationally expensive. Therefore, using TAN through our scaling law can facilitate studying variants of the network architecture while reducing the computational costs.

**Quantity of Data and Privacy Budget**   We now look at how collecting more data affects the tradeoff between privacy and utility. We show that doubling the data (from the same distribution) allows better performance with half the privacy budget. To this end, we train on portions of the CIFAR-10 training set ($N = 50k$) and always report accuracies on the same test set. If we multiply by $\beta$ the quantity of data $N_0$ and keep the same $(B, \sigma, S)$, $q$ (and thus $\eta$), is divided by $\beta$ as well. We divide $\delta$ by $\beta$ for the accounting. We show in Table 5 the effects on $\varepsilon$ and model accuracy.

On the one hand, when using $\varepsilon_{\mathrm{TAN}}$, we can predict the impact on the privacy budget. On the other hand, since the global signal-to-noise ratio $N\eta$ is held constant in all experiments, we expect to extract the same amount of information in each setup; adding more data makes this information richer, which explains the gain in accuracy. We show similar results for ImageNet in Appendix A.

## 5   Conclusions

We argue that the total amount of noise (TAN) is a simple but useful guiding principle to experiment with private training. In particular, we demonstrate that the privacy budget is either a direct function of TAN or can be reduced. We further show that scaling batch size with noise level using TAN allows for ultra-efficient hyper-parameter search and demonstrate the power of this paradigm by establishing a new state of the art for DP training on Imagenet.

### 5.1   Limitations

**Non-private hyper-parameter search.**   We follow the standard practice of not counting hyper-parameter search towards the privacy budget (Li et al., 2021; Anil et al., 2021). Theoretically, each training run should be charged on the overall budget, but in practice it is commonly assumed that the "bandwidth" of hyper-parameters is too small to incur any observable loss of privacy (see also Liu & Talwar (2019) for a theoretically sound way of handling this problem). If available, one can use a similar public dataset (such as ImageNet) to choose hyper-parameters, and then perform only limited runs on the private dataset. Finally, we note that training non-private models might not be possible on sensitive data. In this case, our hyper-parameter transfer process can not be used.

### 5.2   Further work

**Better accounting.**   We believe that the important increase in the privacy budget $\varepsilon$ as the noise level $\sigma$ decreases is a real phenomenon and not an artifact of the analysis. Indeed, DP assumes that the adversary has access to all model updates, as is the case for example in Federated Learning. In such cases, a noise level that is too low is insufficient to hide the presence of individual data points and makes it impossible to obtain reasonable privacy guarantees. In the centralized case however, the adversary does not see intermediate models but only the final result of training. Some works have successfully taken into account this "privacy amplification by iteration" idea (Feldman et al., 2018; Ye & Shokri, 2022) but results are so far limited to convex models.

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

## A    MORE DATA: IMAGENET

We show in Table 6 that similarly to the experiments in CIFAR-10, doubling the training data on ImageNet improves the accuracy while diving $\varepsilon$ by 2. We also demonstrate that our scaling strategy can accurately detect the gain of accuracy. We compare training on half of the ImageNet training set ($N = 600k$) and the entire training set ($N = 1.2M$).

## B    CHOICE OF ARCHITECTURE AND OPTIMIZER

In this section, we give more details about our choice of architecture and optimizer on ImageNet. In particular, we noticed that DP-SGD without momentum is always optimal, even with ViTs, and that NF-ResNets-50 performed the best.

**Architecture.**    When training with DP-SGD, the goal is to find the best possible local minimum within a constrained number of steps $S$, and with noisy gradients. However, architectures and optimizers have been developed to ultimately achieve the best possible final accuracy with normal updates. To illustrate this extremely, we train a Vision Transformer (ViT) (Dosovitskiy et al., 2020) from scratch on ImageNet using DP-SGD. Touvron et al. (2020) have succeeded in achieving SOTA performance in the non-private setting, but with a number of training steps higher than convolution-based architectures. A common explanation is that ViTs have less inductive bias than CNNs: they have to learn them first, and that can be even harder with noisy gradients. And if they are successful, they have lost the budget for gradient steps to learn general properties of images.

We used our scaling strategy (keeping $\eta_{step}$ and $S$ constant) to simulate the DP training with different architectures at low compute, studying noisy training without the burden of DP accounting. The best simulated results were obtained with a NFResNet-50 (Brock et al., 2021b) designed to be fast learners in terms of number of FLOPS. The worst results were obtained with ViTs, and intermediate results with classical ResNets. In Figure 6, we compare different training trajectories of a ViT and a NF-ResNet.

**Optimizer**    Using our simulation scheme, we found that DP-SGD with no momentum and a constant learning rate is the best choice for all architectures. We also tried DP-Adam, DP-AdamW with a wide range of parameters. It is surprising to find that this is the case for ViTs, as without noisy, the Adam type optimizers perform better (Touvron et al., 2020). This highlights the fact that training with DP-SGD is a different paradigm that requires its own tools.

Using TAN allowed us to explore and compare different architectures and optimizers, which would have been computationally impossible in the normal DP training setting at $B = 16384$.

## C    AUGMENTATION MULTIPLICITY

**Augmentation Multiplicity**    (AugMult) was introduced by De et al. (2022) in the context of DP-SGD. The authors average the gradients of different augmentations of the same image before clipping the per-sample gradients, using the following formula (where $\zeta$ is a standard Gaussian variable):

Table 6: Impact of adding more data on ImageNet. The "Simulated Gain" column corresponds to the accuracy gain we observe when simulating at lower compute using our scaling strategy for $B = 256$. The "Gain" column corresponds to the real gain at $B = 16384$.

| Imagenet: $\sigma_{ref} = 2.5$, $B_{ref} = 16384$, $S = 72k$ | | | | | |
|---|---|---|---|---|---|
| N | $\delta$ | $\varepsilon$ | $\varepsilon_{TAN}$ | Gain | Simulated Gain |
| 0.6M | $16.10^{-7}$ | 17.98 | 18.06 | / | / |
| 1.2M | $8.10^{-7}$ | 8.00 | 8.26 | +1.3% | +1.5% |

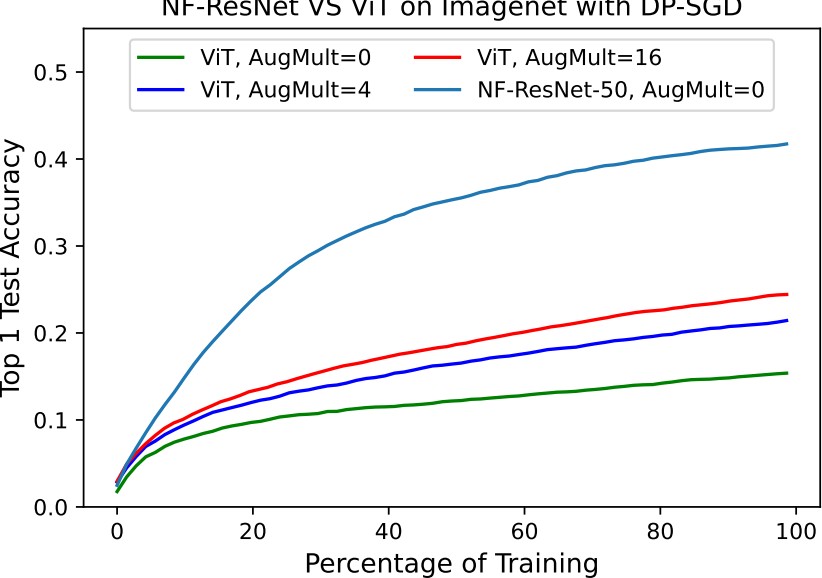

Figure 6: Training a ViT from scratch on ImageNet with DP-SGD. We simulate training with our scaling strategy and $B = 256$. We observe that the accuracies are not as good as for NF-ResNets, and that Augmentation Multiplicity plays a more important role.

$$w^{t+1} = w^{t+1} - \eta_t \left( \frac{1}{B} \sum_{i \in B_t} \frac{1}{C} \text{clip}_C \left( \frac{1}{K} \sum_{j \in K_t} \nabla_j(w^{(t)}) \right) + N \left( 0, \frac{\sigma^2}{B^2} \right) \right) \qquad (5)$$

Compute scales linearly with the AugMult order $K$. Our intuition on the benefits of AugMult is that difficult examples (or examples that fall out of the distribution) become easier when using this augmentation technique. On the other hand, without AugMult, simple examples are learned to be classified early in training, resulting in a gradient close to 0 when used without augmentation. Because we are training for a fixed number of steps, it is a waste of gradient steps (i.e. privacy budget). With AugMult, the network may still be able to learn from these examples. Figure 7 shows the histograms of the norms of the **average over all augmentations for each image** of the per-sample gradients, before clipping and adding noise in equation 5 at different times of training.

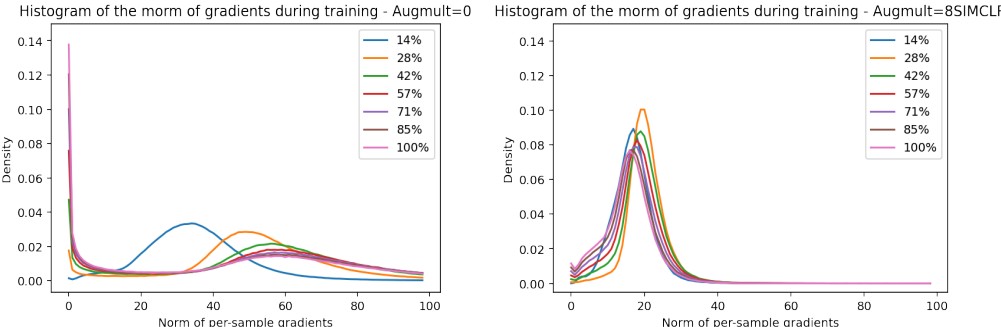

Figure 7: Histograms of the norms of the **average across all augmentations for each image** of the per-sample gradients, before clipping and adding noise. On the left, we see that without augmentation, an increasing number of examples have their gradients going to zero during training. On the right, we see that when using a strong augmentation technique (SimCLR, (Chen et al., 2020)), the gradients are more concentrated during all the training.

