# OpenReview forum: "TAN without a burn: Scaling laws of DP-SGD"
_ICLR.cc/2023/Conference — Submitted to ICLR 2023_

### Official Review · Reviewer_MS1Z · 2022-10-24

**Confidence:** 4
**Correctness:** 4
**Technical Novelty And Significance:** 4
**Empirical Novelty And Significance:** 4
**Recommendation:** 6

**Clarity, Quality, Novelty And Reproducibility:**


Clarity: The paper is overall well-written and clear.

Quality: The observation and the proposed approach are interesting and useful, and the claims are supported by experiments. Overall the quality is good.

Reproducibility: the code of the experiments is not provided.

Novelty: The observation is new and the proposed approach is novel.


**Strength And Weaknesses:**


Strength:

* The observations (batch scaling law, privacy budget computation) in the paper are new and interesting.

* The proposed hyper-parameter tuning process can be very useful in practice, as it saves the computation requirement and therefore speeds up the development process.

Weaknesses:

* Overall, the writing is clear, but there are still some issues that need to be addressed:
    - Equation 2: many of the notations here are not defined, e.g., clip_C, l, theta, C.
    - Figure 4: B_ref, S_ref, \sigma_ref are not defined. I can guess their meaning from the context, but all notations should be defined explicitly in the text.
    - Page 5: it says "Simultaneously doubling the batch size and \sigma has a negligible or small impact on accuracy (Figure 4)". However, in Figure 4(b), the impact on the accuracy is not small. Indeed, this is claimed in the next sentence "Reciprocally, if \sigma > 4, dividing it by 2 simultaneously with halving q is likely to improve performance." These two sentences contradict each other.
    - Table 1: I guess "8,8." should be "8,"?



**Summary Of The Paper:**

Recent work shows that large batch size can give better privacy-utility trade-off in differentia private ML training. However, training with large batch sizes is computationally expensive. The paper observes (empirically) that the privacy budget of differential private ML training depends mostly on the total amount of noise added during training when the noise multiplier \sigma>=2. At the same time, the accuracy of the final results can be predicted by the results on a corresponding hyper-parameter setting in which the batch size is small. Following these observations, the paper proposes to do hyper-parameter tuning for small batch sizes (which is computationally efficient) and then transform the hyper-parameters to the target large batch size. Experiments demonstrate that this approach works well in practice.

**Summary Of The Review:**


Overall, the paper is well-written and the observations and the proposed approach can potentially be very useful for the community. It would be even better if the authors can demonstrate/verify the observation on more network architectures, datasets, and models (e.g., generative models).

---

> ### Author Response · Authors · 2022-11-15
> **Answer to reviewer MS1Z**
>
> We thank the reviewer for their feedback.
>
> - "In Overall, the writing is clear, but there are still some issues that need to be addressed: Equation 2: many of the notations here are not defined, e.g., clip_C, l, theta, C. Figure 4: B_ref, S_ref, \sigma_ref are not defined. I can guess their meaning from the context, but all notations should be defined explicitly in the text. Page 5: it says "Simultaneously doubling the batch size and \sigma has a negligible or small impact on accuracy (Figure 4)". However, in Figure 4(b), the impact on the accuracy is not small. Indeed, this is claimed in the next sentence "Reciprocally, if \sigma > 4, dividing it by 2 simultaneously with halving q is likely to improve performance." These two sentences contradict each other."
>
> We added the definitions for these parameters (changes in blue in the revision of the paper).
> The reviewer is right that our wording was confusing.  We formulate it in a more straightforward manner  in the revision of the paper:  “If $\sigma<2$, simultaneously doubling $q$ and $\sigma$ has a small or negligible negative impact on accuracy (Figure 4) but it can greatly reduce the privacy budget  (Figure 2).
> Reciprocally, halving $\sigma$ and $q$ is slightly beneficial or neutral to the performance (Figure 4) and if $\sigma > 4$,  it keeps the privacy guarantees almost unchanged (Figure 2). It also divides the computational cost by $2$.”
>
> - "Table 1: I guess "8,8." should be "8,"?"
>
> (8, 8.10-7) is a particular value for the couple $(\epsilon, \delta)$
>
> - "Overall, the paper is well-written and the observations and the proposed approach can potentially be very useful for the community. It would be even better if the authors can demonstrate/verify the observation on more network architectures, datasets, and models (e.g., generative models)."
>
> We thank the reviewer for the positive feedback. We used TAN to compare different architectures (see appendix), but indeed not exhaustively. We do believe that trying TAN for other tasks (such as NLP) would be a very interesting next step.

---

> > ### Comment · Reviewer_MS1Z · 2022-11-21
> > **Thank you!**
> >
> > Thank the authors for the explanations. That clarifies my questions.

---

### Official Review · Reviewer_AFbM · 2022-10-24

**Confidence:** 4
**Correctness:** 3
**Technical Novelty And Significance:** 2
**Empirical Novelty And Significance:** 3
**Recommendation:** 6

**Clarity, Quality, Novelty And Reproducibility:**

The paper is well-written, and the finding is novel and useful for tuning DP-SGD.

**Strength And Weaknesses:**

Strength: It studies an important problem of how to scale DP-SGD with hyper-parameter search. The paper provides a neat and simple relationship between the privacy budget and the other parameters, i.e., the noise multiplier, steps and the sampling rate. This relationship motivates a way of scaling the batchsize and the noise multiplier without affecting the privacy accountant.

Weakness:
1. There is no application of using TAN relation in language task, either pretraining or fine-tuning.

2. The reference needs to cover more. Especially, Yu et al. 2021 starts the fine-tuning large language models with differential privacy and low-rank reparametrization.
Yu et al. 2021 Large Scale Private Learning via Low-rank Reparametrization.

**Summary Of The Paper:**



The paper studies how to transfer the hyperparameters of differentially private training based on low-cost search. It shows the privacy budget only depends on the total amount of noise (TAN) injected throughout training. They derives a scaling law for training models with DP-SGD with more than a 100 times reduction in computational budget.

**Summary Of The Review:**

How to tune DP-SGD is both theoretically and practically important for private learning. The paper makes a clear contribution towards tuning DP-SGD in principle. The concluding solution is solid and well-supported. I recommend its acceptance.

\#### After rebuttal and discussion with other reviewers \#####
As discussed with other reviewers, the relation of TAN with privacy accounting has been derived in literature e.g., Lemma 2 and Proposition 3 of Bun et al. 2020. The paper is not fully aware of the literature, which undermines the novelty of the contribution. I change the score to 6.

---

> ### Author Response · Authors · 2022-11-15
> **Answer to reviewer AFbM**
>
> We thank the reviewer for their feedback.
>
> - "There is no application of using TAN relation in language task, either pretraining or fine-tuning."
>
> We note that for fine-tuning, TAN is probably not needed as the computational requirements are smaller. Regarding language tasks, we think that it would indeed be very interesting to explore the use of TAN for language tasks and we leave it to future work.
>
> - The reference needs to cover more. Especially, Yu et al. 2021 starts the fine-tuning large language models with differential privacy and low-rank reparametrization. Yu et al. 2021 Large Scale Private Learning via Low-rank Reparametrization."
>
> We thank the reviewer for this reference that we missed. We added it to the paper.

---

### Official Review · Reviewer_FyxV · 2022-10-25

**Confidence:** 3
**Correctness:** 3
**Technical Novelty And Significance:** 2
**Empirical Novelty And Significance:** 3
**Recommendation:** 6

**Clarity, Quality, Novelty And Reproducibility:**

This paper is well-written. The statements in the paper are clear and concise. The novelty of this paper is good (see the strengths). The code is not provided, and the computation cost of the comparison experiments in this paper seems to be high; therefore the reproducibility is low.

**Strength And Weaknesses:**

Major strengths:
1. The observation of the scaling law is new.
2. The results of hyperparameter tuning improve the state-of-the-art greatly.

Minor strengths:
1. This paper is easy to follow.
2. The visualization of the experimental results is good.
3. Figure 2 seems to deliver the same message as the linear part of Figure 3 (left) in (Li et al, 2021a), but the authors provide more explanations in Section 3.2.

Major weaknesses:
1. The privacy guarantee parameter $\varepsilon_{TAN}$ is an approximation of $\varepsilon_{RDP}$ and it is only a lower bound. For strict privacy guarantees, we need an upper bound.
2. There is no explanation for the reason for the scaling law.

Minor weaknesses:
Some of the statements in this paper are inconsistent with each other.
1. In the left figure of Figure 4, the legend shows that $\sigma_{ref}$ is constant for each line, but the caption says the $\eta_{step}$ is constant. I also wonder why the authors choose to fix different values in Figure 4 for CIFAR-10 and ImageNet. Is it because the results of CIFAR-10 do not look as linear as of ImageNet?
2. In Page 5, Choice of $\sigma$, the authors claimed that if $\sigma>4$, dividing it by 2 with halving q is likely to improve performance, but I cannot find experimental results supporting this.
3. In Page 7, the authors say that they decide not to use the gain from 'testing with augmentations' in Table 1 and Table 4. In Table 2, I guess the Total column is the sum of the other columns for B=128, ..., 1024. Therefore, the last row, B=16384, also has the AugTest (+0.8) in the Total result (6.7%). I guess in Table 4, the test ACC for B=16384, 36.9%, is derived by adding the total improvement 6.7% to the baseline 30.2%. Therefore, I am not sure if the authors forget to remove the improvement from AugTest.
4. In Page 9, in the last paragraph, the authors believe that the exponential increase in the privacy budget $\varepsilon$ as the noise level $\sigma$ decreases. I guess this is from Figure 2, but the x-axis for $\sigma$ is log-scale. Therefore, it is not easy to identify whether there is a linear or exponential relationship between $\varepsilon$ and $\sigma$.

By the way, there are duplicate references (Li et al, 2021a and 2021b).

**Summary Of The Paper:**

The authors proposed to use the total amount of noise (TAN) to determine the privacy budget in Renyi DP. They observe scaling laws with TAN for DP-SGD which can be used to reduce the computational cost for hyper-parameter tuning. By the hyper-parameter tuning, the authors greatly improve the state-of-the-art on neural network training under privacy guarantees.

**Summary Of The Review:**

This paper demonstrates the scaling law between the batch size and the test accuracy in private deep learning. This law and the new concept, TAN, are very useful in improving private learning results by DP-SGD when the computation resources are limited. This law is found in the results of ImageNet and CIFAR-10 experiments, but the reason behind it is still unknown and it may be difficult to reproduce the results since the code is not provided.

---

> ### Author Response · Authors · 2022-11-15
> **Answer to Reviewer FyxV**
>
> We thank the reviewer for their review.
>
> - "The privacy guarantee parameter is an approximation of epsilon  and it is only a lower bound. For strict privacy guarantees, we need an upper bound."
>
> We do not advocate using $\epsilon_{TAN}$ as an upper (nor a lower) bound. Rather we suggest using it as an approximation of the privacy budget that enables quick mental operations (e.g. checking that $\sigma > 2$, scaling batch size and $\sigma$ by the same amount, etc.). To report the actual epsilon accurately, we resort to traditional accounting methods (which provide upper bounds).
>
> - "There is no explanation for the reason for the scaling law."
>
> We observe the scaling laws as an empirical phenomenon. Our running hypothesis is that doubling the batch size leads to more concentrated gradients that are less able to escape from local minimas. Smith et al. (2020) observe the same phenomenon without privacy: when training for a fixed number of steps, performance is also decreasing with the batch-size.
>
> - "Some of the statements in this paper are inconsistent with each other."
>
> We thank the reviewer for these detailed remarks. We submitted a revision of the paper with changes highlighted in blue.
>
> - "In the left figure of Figure 4, the legend shows that $\eta_{step}$  is constant for each line, but the caption says that $\sigma_{ref}$ is constant. I also wonder why the authors choose to fix different values in Figure 4 for CIFAR-10 and ImageNet. Is it because the results of CIFAR-10 do not look as linear as of ImageNet?"
>
> We observe the scaling laws empirically, and demonstrate their validity across 2 data sets with different privacy parameters.
> In the left figure (CIFAR-10), $\eta_{step}$ is indeed constant within each line, and $B_{ref}$  is the same for all curves (4096). We chose to put $\sigma_{ref}$ and not $\eta_{step}$ in the legend, because $\eta_{step}$ is dependent on the data set size N (but not $\sigma$), and because we can compute $\eta_{step}$ easily as $B_{ref} / (\sqrt{2}N\sigma_{ref})$. In the experiments for CIFAR-10, we wanted to show that the scaling law held for small and large noise regimes, and it is valid across a wide range of sigmas (which encompass all practical values). In addition, experiments on CIFAR-10 are computationally cheap; it was therefore possible to have confidence intervals even for big batch sizes.
>
> On ImageNet, we decided to plot the scaling laws at different moments of training: each curve corresponds to a different number of steps of the same training. It is interesting to observe similar scaling laws (approximately the same slope) for all lines.
>
> - "Page 5, Choice of sigma the authors claimed that if $\sigma > 4$, dividing it by 2 with halving q is likely to improve performance, but I cannot find experimental results supporting this."
>
> The reviewer is right that “likely” is a misleading word, so we updated our formulation to “slightly beneficial or neutral” in the article paper. We also added the reference to Figure 4 and Figure 2 more explicitly for better clarity.
>
> - "In Page 7, the authors say that they decide not to use the gain from 'testing with augmentations'  [...]. I am not sure if the authors forget to remove the improvement from AugTest."
>
> The total gain in Table 1 is given with AugTest, as we wanted to show that this testing method improved performance for all batch sizes. Our very final result (39.2%) in Table 4 is given without AugTest, for better comparison with other work, even though it achieved better performance with AugTest. We thank the reviewer for notifying us that the result for B=16384 was incorrectly reported with AugTest. We fixed it.
>
> - "In Page 9, in the last paragraph, the authors believe that the exponential increase in the privacy budget as the noise level sigma decreases. I guess this is from Figure 2, but the x-axis is log-scale. Therefore, it is not easy to identify whether there is a linear or exponential relationship."
>
> We chose this scale to zoom on the exploding phenomenon at small noise sigma and show that it remains constant for larger values of sigma that are used in practice. The reviewer is correct that the x-axis being in log-scale, the relationship is indeed not exponential. Regardless of whether it is linear or exponential, there is a strong increase (and we would expect a constant value). We thank the reviewer for noticing this, we corrected the formulation in the paper.
>
>
> - "there are duplicate references (Li et al, 2021a and 2021b)"
>
> We thank the reviewer for flagging the duplicate reference, it is now corrected in the paper.
>
> - "This law is found in the results of ImageNet and CIFAR-10 experiments, but the reason behind it is still unknown and it may be difficult to reproduce the results since the code is not provided."
>
> See our answer above for the scaling law. We have attached the code in the supplementary for a reference implementation.

---

> > ### Comment · Reviewer_FyxV · 2022-11-17
> > **Thanks for the clarification.**
> >
> > I do not have additional questions.

---

### Official Review · Reviewer_6jSv · 2022-10-26

**Confidence:** 4
**Correctness:** 3
**Technical Novelty And Significance:** 3
**Empirical Novelty And Significance:** 3
**Recommendation:** 6

**Clarity, Quality, Novelty And Reproducibility:**

Clarity:
I found the paper difficult to follow. I think this is because the authors do not clearly acknowledge that, if TAN were an accurate measure of trainability, then test accuracy should not fall as batch size rises (at constant TAN) in figure 1. It is also not always clear which hyper-parameters are extrapolated using the TAN framework, and which are lifted from prior work or tuned in the large batch/low privacy loss regime.

Quality:
I think the underlying quality of the work is quite high (if the clarity can be improved).

Novelty:
This is the first paper I am aware of to provide an explicit toy model predicting the trainability of DP-SGD. Although TAN has some flaws, I think this is a novel and worthwhile research direction.

Reproducibility:
I'm not sure how easy it would be to reproduce the results

**Strength And Weaknesses:**

Strengths:
1) Integrating the constraints of DP-SGD with optimization theory, in order to reliably predict good hyper-parameter choices, is one of the most important problems that needs to be solved in order to make differentially private deep learning practical. I'm pleased to see the authors tackle this difficult and important problem.
2) The metric proposed by the authors is intuitive and it appears to provide some helpful intuition.
3) The authors achieve strong performance when training from scratch on ImageNet with DP-SGD.

Weaknesses:
1) If TAN were an accurate measure of optimizability, then we should expect the train accuracy to be roughly constant at fixed TAN. However in practice, the authors find in figure 1 that the test accuracy decays log-linearly at fixed TAN as the batch size rises. I feel that the current paper is quite confusing because this discrepancy is not explicitly stated anywhere. I'd encourage the authors to make this explicit, and to explore in more detail why this discrepancy arises (eg comparing test vs train performance).

2) As discussed above, Figure 1 shows that TAN over-estimates the performance of large batch sizes. It would be nice to also see a figure at fixed batch size, sweeping the number of steps S, to see whether TAN under/over-estimates the performance of small/large step budgets.

3) As a minor point, note that on some truly sensitive data we cannot train non-private models (even internally). It would be good to acknowledge that the hyper-parameter transfer process described here cannot be used in these cases.

4) For many of the experiments, key hyper-parameters are lifted from prior work (eg step budget, target batch size). Is it not possible to directly infer good settings for these hyper-parameters using the TAN framework?

5) The paper essentially only considers ImageNet (with some very limited experiments on CIFAR-10). Since very few groups have run experiments at this scale it is not clear how challenging the baseline from De et al. is in practice.



**Summary Of The Paper:**

The authors argue that our ability to optimize a model trained with DP-SGD is governed by a quantity they call TAN (the total amount of noise), specifically $\Sigma^2 = 2 \sigma^2 N^2/(B^2 S)$, where $\Sigma$ is the TAN, $\sigma$ is the scale of the added Gaussian noise, $N$ is the dataset size, $B$ is the batch size and $S$ is the number of updates.

Additionally, they show empirically that when $\sigma \gtrsim 2$, the privacy parameter $\epsilon$ becomes a simple function of $\Sigma$ and $\delta$. Since it is standard to set $\delta \approx 1/N$, in practice this means that the privacy is determined solely by $\Sigma$, the TAN. Meanwhile, the privacy loss $\epsilon$ is very large when $\sigma \ll 2$.

The authors propose to exploit this observation by tuning hyper-parameters with small $\sigma$, small batch size $B$ and large privacy loss $\epsilon$, before extrapolating those hyper-parameters to $\sigma \gtrsim 2$ while increasing the batch size to ensure that the TAN $\Sigma$ is constant. This significantly reduces the privacy loss $\epsilon$ however it also significantly increases the compute cost of training. Extrapolating hyper-parameters from cheap non-private training runs to expensive runs with tight privacy guarantees reduces the total computation required to achieve strong performance with DP.

**Summary Of The Review:**

On the positive side, the authors identify an important and difficult problem, and provide some useful insights. However on the downside TAN is not a reliable measure of performance in practice, and the authors only achieve strong results on one dataset so it is not clear how reliable the methodology is.

I have scored the paper weak accept, however I would encourage the authors to improve the clarity of the writing and to empirically explore in more detail how test accuracy depends on (batch size, step budget, added noise) under the constraint of constant TAN.

---

> ### Author Response · Authors · 2022-11-15
> **Answer to Reviewer 6jSv**
>
> We thank the reviewer for their feedback.
>
> -  "If TAN were an accurate measure of optimizability, then we should expect the train accuracy to be roughly constant at fixed TAN. However in practice, the authors find in figure 1 that the test accuracy decays log-linearly at fixed TAN as the batch size rises. I feel that the current paper is quite confusing because this discrepancy is not explicitly stated anywhere. I'd encourage the authors to make this explicit, and to explore in more detail why this discrepancy arises (eg comparing test vs train performance)."
>
> We will clarify this in the writing. Even though we expected TAN to predict performance, we introduce it initially as a way to quantify privacy.  Regarding the underlying reason for the scaling laws, our hypothesis is that doubling the batch size leads to more concentrated gradients that are less able to escape from local minimas (even with DP noise). Smith et al. (2020), show that without privacy, when training for a fixed number of steps, performance is also decreasing with the batch size. Finally, on both ImageNet and CIFAR-10, the training accuracies were very similar to the test accuracies throughout training.
>
> - "As discussed above, Figure 1 shows that TAN over-estimates the performance of large batch sizes. It would be nice to also see a figure at fixed batch size, sweeping the number of steps S, to see whether TAN under/over-estimates the performance of small/large step budgets."
>
> The reviewer is correct that it would be interesting to gain more general insights into the relationship between TAN and performance. In the paper, we chose to vary batch size with sigma as it allows to have a constant number of steps (and a constant noise per step). Given that neural networks are sensitive to the number of steps, we chose this approach to avoid confounding factors.
> We discuss the number of steps in Section 4.2.2, where we use TAN to choose the privacy parameters optimally.
>
> - "As a minor point, note that on some truly sensitive data we cannot train non-private models (even internally). It would be good to acknowledge that the hyper-parameter transfer process described here cannot be used in these cases."
>
> This is a good point; we added this acknowledgment in our limitations.
>
> - "For many of the experiments, key hyper-parameters are lifted from prior work (eg step budget, target batch size). Is it not possible to directly infer good settings for these hyper-parameters using the TAN framework?"
>
> We decoupled privacy hyper-parameters (HPs) from non-privacy HPs in our experiments. In Section 3, we use TAN to find better non-privacy HPs keeping the privacy HPs (target batch size, step budget and sigma) fixed from prior work. However, in Section 4.2, we directly use TAN to optimally choose these privacy HPs (which further improves performance by 3 points) and that constitutes our state-of-the-art run.
>
> - "The paper essentially only considers ImageNet (with some very limited experiments on CIFAR-10). Since very few groups have run experiments at this scale it is not clear how challenging the baseline from De et al. is in practice."
>
> Experiments on ImageNet do indeed require experimental resources. One of the goals of our paper is precisely to decrease that requirement, and we hope that our method makes Imagenet-scale experiments with privacy more accessible.
> Regarding the results from De et al., we note that the performance on ImageNet with privacy was very low before (e.g., Kurakin et al. 2022 report 8% under epsilon=8), and De et al. made a significant improvement, lifting the state of the art to 32%.
>
> - "I would encourage the authors to improve the clarity of the writing and to empirically explore in more detail how test accuracy depends on (batch size, step budget, added noise) under the constraint of constant TAN."
>
> We thank the reviewer for these suggestions. We have refactored some parts of the paper and will submit a revision with changes highlighted in blue.  Other dependencies would indeed  be interesting to explore. We will be performing more experiments to explore the TAN-performance relation.
>
> - "Reproducibility: I'm not sure how easy it would be to reproduce the results"
>
> We have attached the code in the supplementary for a reference implementation.

---

> > ### Comment · Reviewer_6jSv · 2022-11-23
> > **thanks for your response**
> >
> > Thanks for your helpful response. I'll reply to some of the points below:
> >
> > 1) Thank you for agreeing to clarify that, from first principles, one would have expected the performance at constant TAN to be constant; I think this could make the paper a lot easier to follow.
> >
> > I agree that a likely explanation for why performance is not constant at constant TAN is the generalization benefit of small batch sizes (as seen in non-private training). I would like to emphasize that this is very surprising though and worthy of further study. I would have expected that the noise in DP-SGD, which already largely prevents overfitting, would be large enough to overwhelm any benefit of small batch training. One possible experiment here would be to check if TAN becomes a more reliable measure when $\epsilon$ is small?
> >
> > 2) I would still encourage the authors to investigate whether TAN is a reliable predictor of performance at constant batch size and variable step budget. I think this is important to establish how reliable TAN is in practice or whether it only holds when the step budget is fixed. I also think this could help shed light on the questions above.
> >
> > 3) I am still concerned that the baseline from De et al. may not in practice be very strong. They are effectively the only group to have studied this task (given that the previous baseline was so weak), and additionally it is only a minor part of their paper. They explicitly state that they ran a "very limited sweep" of hyper-parameters for this task. I think the paper would be a lot stronger if it also included strong results on more commonly used benchmarks (eg CIFAR-10), even if only to show that the TAN method quickly finds hyper-parameters close to SOTA.

---

> > > ### Author Response · Authors · 2022-11-29
> > > **Answer to Reviewer 6jSv**
> > >
> > > We thank the reviewer for their precious and constructive feedback.
> > >
> > > - "I agree that a likely explanation for why performance is not constant at constant TAN is the generalization benefit of small batch sizes (as seen in non-private training). I would like to emphasize that this is very surprising though and worthy of further study. I would have expected that the noise in DP-SGD, which already largely prevents overfitting, would be large enough to overwhelm any benefit of small batch training. One possible experiment here would be to check if TAN becomes a more reliable measure when ϵ is small?"
> > >
> > > We had the same intuition when we first thought of using TAN that the artificial Gaussian noise would largely cancel out the natural SGD noise. We agree that this is an interesting direction for future work.
> > >
> > > On CIFAR-10 (Figure 4 left), we showcase the scaling laws for different noise levels and observe the same behavior in the experiments with large and small noise (therefore large and small epsilon). The minimum value of $\sigma_{ref}=0.5$, with $B_{ref}=4096$, was chosen such that the ratio  $\sigma$/B is close to the one we used on ImageNet ($\sigma_{ref}$=2.5, $B_{ref}$=16384), in order to have the same noise magnitude. On CIFAR-10, the performance is almost constant (slope=0), while it is clearly not the case for the same level of signal-to-noise per step on ImageNet.
> > >
> > > Similarly, the reviewer is right that it would be interesting to progressively increase sigma on ImageNet  (therefore decreasing epsilon) to study how the laws change. We expect an increasing slope (artificial noise canceling out the natural noise) and a decreasing intercept (because of noisier gradients).
> > >
> > > - "I would still encourage the authors to investigate whether TAN is a reliable predictor of performance at constant batch size and variable step budget. I think this is important to establish how reliable TAN is in practice or whether it only holds when the step budget is fixed. I also think this could help shed light on the questions above."
> > >
> > > The reviewer suggests to vary S and $\sigma$ but to keep q constant, overall keeping TAN constant. We emphasize that our initial goal with TAN was to be more computationally efficient while minimizing changes to the training process (same number of steps and same signal-to-noise ratio per step). However, we agree that varying the number of steps would indeed help to gain more insight into the relationship between TAN and performance outside of computational considerations. We will update the paper to clarify that we only explore TAN at a fixed number of steps.
> > >
> > > - "I am still concerned that the baseline from De et al. may not in practice be very strong. They are effectively the only group to have studied this task (given that the previous baseline was so weak), and additionally it is only a minor part of their paper. They explicitly state that they ran a "very limited sweep" of hyper-parameters for this task. I think the paper would be a lot stronger if it also included strong results on more commonly used benchmarks (eg CIFAR-10), even if only to show that the TAN method quickly finds hyper-parameters close to SOTA."
> > >
> > > We argue that the baseline from De et al. is competitive if the computational cost of each training run is high. We view our work mainly as a general method to drastically reduce the computational cost of parameter sweeps, and the new state of the art on Imagenet as an instantiation of our method. We think that our new state of the art is strong, but we hope that the TAN method allows other groups to propose new ideas and test them with a reduced computational cost.
> > >
> > > We would also like to highlight our ablation on CIFAR-10 (Section 4.3/Figure 5) in more detail. We show that a small batch size simulation allows to identify the importance of the order between group norm and ReLU within a WideResNet. TAN can thus be used to optimally choose this hyper parameter (the order between the two) on CIFAR-10.
> > >
> > > The reviewer is right that performing an exhaustive sweep would help emphasize the usefulness of TAN, even though it would likely not result in any improvement over the baseline because the computational barrier is already low. We will perform these experiments for the next revision of our paper.

---

### Author Response · Authors · 2022-11-15
**Response to all**

We thank the reviewers for their insightful comments and questions. We have updated a revision of the paper to take these comments into account (changes made in blue), and added anonymized code for reproduction.

Main changes:
- Fixed typos
- Added forgotten definitions
- Added missing reference
- Added limitation
- Attached code for reproduction

---

### Decision · Program_Chairs · 2023-01-20

**Decision:**

Reject

**Justification For Why Not Higher Score:**

See the discussion above.

**Justification For Why Not Lower Score:**

N/A

**Metareview: Summary, Strengths And Weaknesses:**

The paper studies the hyperparameter choices of DP-SGD, arguably the most popular method for deep learning with differential privacy.  The main finding of the paper is the claimed discovery of the so-called "Total Amount of Noise"  (abbrv TAN), denoted by $q^2S / (2 \sigma^2)$ where $q$ is the sampling probability for the minibatch,  $S$ is the number of iterations and $\sigma$ is the "noise multiplier" (the ratio between the standard deviation of the noise and the clipping threshold).

The paper states that when \sigma > 2, TAN provides a good approximation of the RDP parameter in the useful ranges for obtaining (eps,delta)-DP.   The paper then used TAN to inform on the hyperparameter choices for DP-SGD. It recommended a particular configuration of the hyperparameters (fix S, choose $\sigma \in [2,4]$, and decide on q according to a given privacy budget).  Experiments show that the better hyperparameter choices (smaller batch size, smaller amount of noise while keeping TAN the same), provide stronger results in training ImageNet from scratch using DP-SGD.

The strengths of the paper include:
- new SOTA for imagenet with DP
- an intuitive explanation of TAN  (and an informal discussion on how it is related to "effective noise" from Li et al. 2021)
- interesting experimental results that demonstrate that test accuracy reduces linearly w.r.t. the log of BatchSize.

The weakness of the paper include (some of these come up in the AC-reviewer discussion):
- TAN, and the observed relationship between TAN and privacy loss in the regime of interest is not new, and in fact well-known.
- TAN does not explain the interesting linear scaling between the  "test accuracy and the log of BatchSize"; and the explanation that appeals to better "generalization" is ad hoc and is unlikely to be the reason.
- The paper seems to have missed well-known convergence results in DP-SGD. The claimed new insight on hyperaparameter choices appears to be directly implied by theory.

More details of the above are provided in the summary of AC-reviewer meeting below.







**Summary Of Ac-Reviewer Meeting:**

The meeting focused on the discussion on the novelty of the work, after the AC pointed out a few key references the reviewers (and the authors) overlooked.

While the authors acknowledged that the connection between TAN and privacy losses are well-known from the DP-accounting literature with Renyi Differential Privacy,  the authors claimed that TAN is a useful empirical rule-of-thumb that is simpler than the rigorous bounds from RDP.  However, the same simple form in the restricted regimes of interest has also been well-known. The AC wrote:

* "TAN is not proposed in this paper. It is proportional to the linear phase (in alpha) of the RDP parameters studied by Abadi et al. (2016), Wang et al.. (2018), Mironov et al. (2019). A more explicit form was written in Proposition 3 of the truncated CDP paper by Bun et al: https://projects.iq.harvard.edu/files/privacytools/files/bun_mark_composable_.pdf It is merely a renamed and less rigorous version of the truncated CDP parameter of DP-SGD, known to describe the privacy parameter of the algorithm in the regime being considered. The authors did not appear to have cited the tCDP paper. "

The discussion then considered the novelty to be potentially applying the rule empirically as a simple rule-of-thumb.  But that has been done too (arguably more rigorously and more thoroughly) in the paper "Deep Learning with Gaussian Differential Privacy": https://arxiv.org/abs/1911.11607
See the last paragraph of Page 8 and the beginning of Page 9.  The GDP parameter (after appealing to CLT) is almost the same as what the authors proposed in the regime of interest (i.e., when $\sigma>4$  $e^{1/\sigma^2}-1 \approx  \frac{1}{\sigma^2}$  )

The AC also pointed out that despite the intention to integrate TAN with optimization theory, the authors did not try to discuss what the convergence theory of DP-SGD (effectively the classical theory of SGD) would suggest in contrast to the proposed hyperparameter regime.  Specifically, the AC wrote:

"The paper does not contain any theoretical characterization of their scaling rule, even though the convergence theory for DP-SGD in all parameters is well known for both convex, strongly convex, and non-convex (stationary point convergence) settings. A lot of the scaling discussion in this paper can be directly read off from the theory.  It is a bit unsatisfactory that the authors did not try to relate to the existing theory with their experimental results for consistency (or other otherwise!)"

As a concrete example, the standard convergence analysis of DP-SGD (almost identical to Bassily et al., 2014) in the regime of sigma > 2 gives the following bound (for the smooth convex DP-ERM problem)

>$$
\text{Suboptimality} \lesssim \frac{\|x_1-x^*\|^2}{ S \eta} + \eta (\frac{d \sigma^2C^2}{q^2} + \frac{n \beta}{q})
$$

where $x_1$ is the initial weights, $x*$ is the optimal weights, $n$ is the number of data points and $\eta$ is the learning rate (assuming the per-example loss is 1-Lipschitz, and $\beta$-smooth).   The LHS is the difference between the sum of the loss functions over $n$ data points at the output of DP-SGD minus that at the (nonprivate) empirical risk minimizer.

The above formula can be minimized by choosing the learning rate parameter $\eta$ optimally then the second term would be parameterized only by TAN  (or the rho-tCDP parameter, or the GDP parameter).  One can impose any other restrictions, e.g.,fixing S as the authors did and work out the relationship between different choices.

Of course, the LHS of the bound is not test error in 0-1 loss, so it doesn't directly cover the empirical finding in this paper. Indeed, the AC and reviewers all find that the linear scaling between test accuracy and the log-batch size when fixing the privacy budget quite an intriguing observation, but could not figure out how TAN has anything to do with it.

The conclusion of the discussion ends with two choices:

- Accepting the paper to ICLR and hope the authors would rewrite some part of the paper by acknowledging the new references and clearly discussing the novel contributions.

- Requesting the authors to do a major revision and resubmit to the next conference.

The AC decides that the latter is more appropriate given the difficulty in completing such a revision within a short window. This must be very disappointing news to the authors, but the committee hopes the authors could understand the comments and take the chance to make the paper even stronger. Ultimately, we believe the paper will have the potential to be highly impactful.